# Exploit Reward Shifting in Value-Based Deep-RL: Optimistic Curiosity-Based Exploration and Conservative Exploitation via Linear Reward Shaping

Hao Sun[1][*]    Lei Han[2]    Rui Yang[3]    Xiaoteng Ma[4]    Jian Guo[5]    Bolei Zhou[6]

## Abstract

In this work, we study the simple yet universally applicable case of reward shaping in value-based Deep Reinforcement Learning (DRL). We show that reward shifting in the form of a linear transformation is equivalent to changing the initialization of the $Q$-function in neural approximation. Based on such an equivalence, we bring the key insight that a positive reward shifting leads to conservative exploitation, while a negative reward shifting leads to curiosity-driven exploration. Accordingly, conservative exploitation improves offline RL value estimation, and optimistic value estimation improves exploration for online RL. We validate our insight on a range of RL tasks and show its improvement over baselines: (1) In offline RL, the conservative exploitation leads to improved performance based on off-the-shelf algorithms; (2) In online continuous control, multiple value functions with different shifting constants can be used to tackle the exploration-exploitation dilemma for better sample efficiency; (3) In discrete control tasks, a negative reward shifting yields an improvement over the curiosity-based exploration method. https://sites.google.com/view/rewardshaping

## 1 Introduction

While reward shaping is a well-established practice in reinforcement learning and has a long-standing history [1, 2], specifying a certain reward to incentivize the learning agent requires domain knowledge and a thorough understanding of the task [3–6]. Even with careful design and tuning, learning with a shaped reward that intends to accelerate learning may on the contrary hinder the learning performance by incurring the sub-optimal behaviors of the agent [7, 8]. Although Ng et al. [9] theoretically points out that optimal policy remains unchanged under certain form of reward transformation, and in the later work of Wiewiora et al. [10] a framework is proposed to guide policies with prior knowledge under tabular setting, how reward shifting accommodates recent Deep Reinforcement Learning (DRL) algorithms remains much less explored.

In this work, we study a special linear transformation, which is the simplest form of reward shaping,

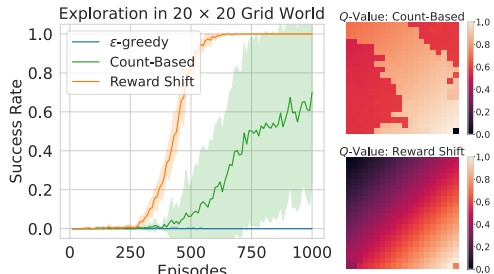

Figure 1: Our work is inspired by the observation that reward shifting remarkably helps exploration and outperforms count-based exploration in $Q$-learning (left). Reward shifting does not change the primal optimal $Q$-value landscape, and is able to learn a near-optimal $Q$-value (right). While count-based exploration suffers from the curse of dimensionality, reward shifting can be seamlessly applied to high-dim tasks including continuous control.

[*]hs789@cam.ac.uk. [1]University of Cambridge; [2] Tencent RoboticsX; [3]Hong Kong University of Science and Technology; [4] Tsinghua University; [5] IDEA; [6] University of California, Los Angeles

36th Conference on Neural Information Processing Systems (NeurIPS 2022).

in value-based DRL [11–14]. We start with understanding how such a specific kind of reward shaping works in value-based DRL function approximations and show that reward shifting is equivalent to different $Q$-value initialization, extending previous discovery of [10] to the function approximation setting. Figure 1 showcases reward shifting benefits exploration in a maze game and outperforms count-based and $\epsilon$-greedy exploration and learns near optimal value function.

Based on such an equivalence, we introduce the key insight of this work:

> *A positive reward shifting leads to conservative exploitation, and a negative reward shifting leads to (a new type [2] of) curiosity-driven exploration.*

We demonstrate the application of such an insight in **three Deep-RL scenarios: (S1) for offline RL**, we show that conservative exploitation induced by reward shifting improves learning performance of off-the-shelf algorithms; **(S2) for online RL setting**, we show multiple value functions with different reward shifting constants can be used to trade-off between exploration and exploitation, thus improve learning efficiency; **(S3) for curiosity-driven exploration,** we introduce a simple yet crucial adaptation on a prevailing curiosity-based exploration algorithm, the Random Network Distillation [15], making it compatible with value-based DRL algorithms. Experiments on a diverse set of tasks including continuous and discrete action space control show our method brings substantial improvement over baselines.

**Our contributions** can be summarized as follows:

1. Analytically, we introduce the key insight that reward shifting is equivalent to diversified $Q$-value network initialization in value-based DRL, which can be applied to curiosity-driven exploration and conservative exploitation;
2. Practically, we instantiate the key insight to three different scenarios, namely the offline conservative exploitation, sample-efficient continuous control, and curiosity-driven exploration, to contrast the generality of reward shifting;
3. Empirically, we demonstrate the effectiveness of the proposed method integrated with multiple off-the-shelf baselines on both continuous and discrete control tasks.

## 2 Preliminaries and Related Work

### 2.1 Online RL

We follow a standard MDP formulation in the online RL setting, i.e., $\mathcal{M} = \{\mathcal{S}, \mathcal{A}, \mathcal{T}, \mathcal{R}, \rho_0, \gamma, T\}$, where $\mathcal{S} \subset \mathbb{R}^d$ denotes the $d$-dim state space, $\mathcal{A}$ is the action space (note for discrete action space $|\mathcal{A}| < \infty$ and for continuous control $|\mathcal{A}| = \infty$), we consider a deterministic transition dynamics $\mathcal{T} : \mathcal{S} \times \mathcal{A} \mapsto \mathcal{S}$ and deterministic reward function $\mathcal{R} : \mathcal{S} \times \mathcal{A} \mapsto \mathbb{R}$. $\rho_0 = p(s_0) \in \Delta(\mathcal{S})$ denotes the initial state distribution. $\gamma$ is the discount factor and $T$ is the episodic decision horizon. Online RL considers the problem of learning a policy $\pi \in \Pi : \mathcal{S} \mapsto \Delta(\mathcal{A})$ (or $\pi \in \Pi : \mathcal{S} \mapsto \mathcal{A}$ with a deterministic policy class), such that the expected cumulative reward $\mathbb{E}_{a_t \sim \pi, s_{t+1} \sim \mathcal{T}, s_0 \sim \rho_0} \sum_{t=0}^{T} \gamma^t r_t(s_t, a_t)$ in the Markovian decision process is maximized. In online RL setting, an agent learns through trials and errors [11], through either on-policy [16–18] or off-policy paradigm [12–14, 19–21]. In this work, we focus on the off-policy value-based methods which are in general more sample efficient. Specifically, our discussions assume the policy learning is based on a learned $Q$-value function that approximates the cumulative reward an agent can gain in the following part of an episode. Formally, $Q(s_t, a_t) = \mathbb{E}_{\pi, \mathcal{T}} \sum_{\tau=t}^{T} \gamma^t r(s_\tau, a_\tau)$, and can be estimated by propagating the Bellman operator $\mathbb{B}Q(s, a) = r(s, a) + \gamma \mathbb{E}Q(s', a')$. For value-based methods, the (soft-)optimal policy is then produced by

$$\pi_\alpha^*(a|s) = \frac{\exp \frac{1}{\alpha} Q^*(s, a)}{\sum_{a'} \exp \frac{1}{\alpha} Q^*(s, a')}, \tag{1}$$

where $Q^*$ is the optimal $Q$-value function. We can also set the temperature parameter close to 0 to have the deterministic policy class. Simplifying the notion we have $\pi^*(s) = \arg\max_a Q^*(s, a)$. Algorithms like DPG [22] can be used to address the intractable analytical argmax issue that arises in continuous action space. We choose to develop our work on top of prevailing baseline algorithms of

---

[2]The novelty this new type of curiosity is presented in Section 4.3.2

Table 1: Reward shifting is flexible to be plugged into both online and offline RL algorithms to guarantee conservative exploitation or pursue optimistic exploration. It covers both discrete and continuous control tasks, with only a little additional computational expense. Moreover, the optimal policy learned with shifted reward is not biased.

| Covered Topics | Related Work | Plug-in | Online | Offline | Discrete | Continuous | Unbiased | Examples |
|---|---|---|---|---|---|---|---|---|
| Exploration | Curiosity | ✓ | ✓ | · | ✓ | · | · | Burda et al. [15] |
| | Ensemble | · | ✓ | · | ✓ | · | ✓ | Osband et al. [25] |
| | Initialization | · | ✓ | · | ✓ | · | ✓ | Rashid et al. [26] |
| | Optimism | · | ✓ | · | · | ✓ | · | Ciosek et al. [27] |
| Exploitation | Conservatism | · | · | ✓ | ✓ | ✓ | · | Bharadhwaj et al. [24] |
| | Policy Constraints | ✓ | · | ✓ | ✓ | ✓ | ✓ | Fujimoto et al. [23] |
| | Reward Shifting | ✓ | ✓ | ✓ | ✓ | ✓ | ✓ | Ours |

DQN [13], BCQ [23], CQL [24] and TD3 [14] as a minimal example to isolate the source of gains. It should be straightforward to extend the method on top of other learning algorithms.

## 2.2 Exploration and the Curiosity-Driven Methods

One of the most important issues in online RL is the exploration-exploitation dilemma [11] that the agent must simultaneously learn to exploit its accumulated knowledge on the task and explore new states and actions. Previous works address the exploration problem from various perspectives: for discrete action space tasks, count-based methods like [28–30] are proposed to motivate the revisiting of under-explored state-action pairs. Specifically, Choshen et al. [31] extended the idea into general settings by constructing an additional MDP for Exploration-value estimation, as a generalized counter for count-based exploration. To boost exploration in continuous state tasks, curiosity-driven methods are investigated by Houthooft et al. [32], Pathak et al. [33], Burda et al. [34, 15], where variety of intrinsic rewards are designed as supplementary to the primal task reward for better exploration. Self-imitate approaches like Oh et al. [35], Ecoffet et al. [36], Sun et al. [37] repeat success trajectories but require extra assumptions on the environment.

DORA [31] constructed an additional MDP to estimate the Exploration-value as a generalised counter for count-based exploration, yet those count-based methods are orthogonal to reward shifting: in intrinsic reward methods, an agent must **first experience** a new $(s, a)$ pair before receiving a high intrinsic reward — this is extremely hard with an arg-max style policy. On the other hand, with optimistic initialization, the rarely-visited $(s, a)$ pairs will naturally have higher $Q$-values **before experiencing** it — as the frequently-visited pairs have updated their values with a negatively shifted reward. From such a perspective, reward shifting not only works by itself motivates exploratory behaviors, but can also be seamlessly plugged into intrinsic reward methods to **encourage the first visitation** of new states.

The works of DIAYN and DADS [38, 39] show that various skills can be developed even without the primal extrinsic reward. For continuous control tasks, OAC [27] improves the SAC [20] with informative action space noise based on the optimism in face of uncertainty (OFU) [40–43]. GAC [44] addresses the exploration issue with a richer functional class for the policy.

In the work of Rashid et al. [26], the problematic pessimistic initialization is addressed for better exploration, yet the work focuses on specific settings of tabular and discrete control exploration. In the work of Osband et al. [45, 25], ensemble models with diverse initialization and randomized priors are used to resemble the insight of bootstrap sampling and facilitate better value estimation, yet those methods are only applicable to discrete control tasks. Noted that although the reward shifting can be regarded as a special case of these random priors, it can be distinguished by not changing the optimal $Q$-value, and its flexibility to be plugged into both continuous and discrete control algorithms.

Random Network Distillation (RND) [15] defines the intrinsic reward as the output difference between a fixed neural network $\phi_1$ and a trainable network $\phi_2$ given state-actions as the inputs. e.g.,

$$r_{\text{int}}(s, a) = |\phi_2(s, a) - \phi_1(s, a)|, \tag{2}$$

where both networks are activated by a sigmoid function. After optimizing the learnable $\phi_2$ to approximate $\phi_1$ with seen $(s, a)$ pairs [3], the value of $r_{\text{int}}(s, a)$ will decay to 0 for such state-action pairs that are frequently visited but remain high for thoase are seldom visited.

---

[3]Or computed with only states, i.e., $r_{\text{int}}(s) = |\phi_2(s) - \phi_1(s)|$.

In this work, we show that exploratory behavior can be achieved simply by shifting the reward function with a constant. Thus our method is orthogonal to those previous approaches in the sense that our intrinsic exploration behavior is motivated by high function approximation error in the under-explored state-action pairs. We demonstrate such insight by showing that RND in its original design is not suitable for value-based methods in developing exploratory behaviors, but integrating RND with a shifted reward function remarkably improves the learning performance.

## 2.3 Offline RL

The offline RL, also known as batch-RL, focuses on the problems where interaction with the environment is impossible, and the policy can only be optimized based on the logged dataset. In those tasks, a fixed buffer $\mathcal{B} = \{s_i, a_i, r_i, s_i'\}_{i=[N]}$, collected from some unknown behavior policy $\pi_\beta$, is provided. In general, such a dataset can either be generated by rolling out an expert that generates high-quality solutions to the task [23, 46, 47] or a non-expert that executes sub-optimal behaviors [47–51] or be a mixture of both [24]. As the agent in the offline RL setting can not correct its potentially biased knowledge through interactions, the most important issue is to address the extrapolation error [23] induced by distributional mismatch [49]. To address such an issue, a series of algorithms optimize the policy learning under the constraint of distributional similarity [48, 49, 52, 53].

Bharadhwaj et al. [24] propose CQL to solve the offline RL tasks with a conservative value estimation. Specifically, CQL learns the $Q$-value estimation by jointly maximizing the $Q$-values of actions sampled from the behavior offline dataset and minimizing the $Q$-values of actions sampled with pre-defined prior distributions (e.g., uniform distribution over the action space). As we will show in this work, an alternative approach to have a lower bound for the optimal $Q$-value function is to use an appropriately shifted reward function. This idea leads to the direct application of our proposed framework in the offline setting. In general, reward shift can be plugged in many distribution-matching offline-RL algorithms [23, 48, 49, 52] to further improve the performance with conservative $Q$-value estimation.

Table 1 contextualizes reward shifting with respect to related works we discussed above.

## 3 A Motivating Example

We start with two intuitive remarks and a "counter-intuitive" motivating example in this section before introducing our method.

**Remark 1.** Given an MDP $\mathcal{M} = \{\mathcal{S}, \mathcal{A}, \mathcal{T}, \mathcal{R}, \rho_0, \gamma\}$, where $|\mathcal{A}| < \infty$, scaling the reward function with linear transformation, i.e., $\mathcal{R}_{k,b} = k \cdot \mathcal{R} + b, \forall k > 0, b \in \mathbb{R}$, such that $r_t' = kr_t + b \in \mathcal{R}_{k,b}$, does not change the optimal policy induced by the corresponding value function $Q_{k,b}^*(s,a) := \sum_t \gamma^t r_t'$:

$$\pi^*(s) = \arg\max_{a \in \mathcal{A}} Q_{k,b}^*(s,a) = \arg\max_{a \in \mathcal{A}} kQ^*(s,a) + \frac{b}{1-\gamma} = \arg\max_{a \in \mathcal{A}} Q^*(s,a), \quad (3)$$

**Remark 2.** When $|\mathcal{A}| = \infty$, scaling the reward function with linear transformation does not change the optimal policy induced by deterministic policy gradient [22], given proper learning rate $\eta' = \eta/k$:

$$\nabla_\theta J(\mu_\theta) = \mathbb{E}_{s_t}[\nabla_a Q^*(s_t, a_t)|_{a_t = \mu_\theta(s_t)} \nabla_\theta \mu_\theta(s_t)] = \mathbb{E}_{s_t}[\nabla_a Q_{k,b}^*(s_t, a_t)|_{a_t = \mu_\theta(s_t)} \nabla_\theta \mu_\theta(s_t)]/k, \quad (4)$$

Remark 1 and Remark 2 declare the fact that constant reward shifting does not affect the optimal policy induced by the optimal $Q$-value function calculated by the shifted reward. However, Figure 1 presents "counter-intuitive" results in a demonstrative exploration task of Grid World. In this task, an agent located at the upper left corner of a map needs to explore without reward and reach the goal point located at the lower right corner. A non-trivial reward of $+1$ will be assigned only when the goal is reached. We report learning curves with regard to the episodic success rate in reaching the goal point and the learned $Q$-values. In this toy example, we find a negative reward shifting remarkably boosts the learning efficiency of $Q$-learning, and surpasses the conventional count-based method for exploration. Moreover, Remark 1 is empirically verified with such a toy example: reward shifting does not change the optimal $Q$-value as well as its induced policy, but **it does change the learning dynamics** and accelerates the discovery of (near-)optimal policy.

In the following of this work, we study the effect of varying constant bias $b$ and fix the scaling factor $k = 1$ to avoid trivial discussions on the choices of learning rate — there should be no surprise that

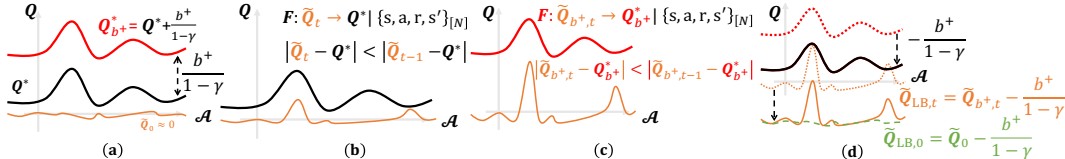

Figure 2: Illustrative figure for conservative exploitation, with a positive constant bias added to the reward function. We use **Black** lines to denote the original value function, and use **Red** lines to denote the shifted value function with the constant reward shift. **Orange** lines are used to denote the function approximation during different learning stages and **Green** line shows the equivalent **pessimistic initialization**. **(a)** shifting the reward function with a positive bias term $b^+$ leads to an uniformly increased $Q$-value function, namely $Q_{b^+}^* = Q^* + \frac{b^+}{1-\gamma}$, during learning, a neural network estimator $\tilde{Q}$ initialized with $\tilde{Q}_0 \approx 0$ is optimized to approximate the $Q$-value functions (e.g., through TD or MC estimation). **(b)** for any value-based RL algorithm, the value optimization step can be regarded as a function $F$ that minimizes the difference between the estimated $Q$-value function $\tilde{Q}_t$ and the optimal one $Q^*$, given the interaction experience with the environment (e.g., a replay buffer $\mathcal{B}$ for off-policy methods). Note $\tilde{Q}_t$ approximating $Q^*$ better than $\tilde{Q}_{t-1}$ holds in expectation as long as the RL algorithm is designed to approximate the optimal value function. **(c)** similarly, the optimizer given the same interactive experience (e.g., replay buffer $\mathcal{B}$) will learn to minimize the difference between $Q$-value function $\tilde{Q}_{b^+,t}$ and the optimal one $Q_{b^+}^*$, after re-labeling the rewards in the buffer by $r' = r + b^+$. **(d)** according to Remark 2, the optimization conducted in (c) is equivalent to (b) with the neural network $Q$-value estimator initialized as $\tilde{Q}_{\mathrm{LB},0} \approx \tilde{Q}_0 - \frac{b^+}{1-\gamma}$, rather than $\tilde{Q}_0 \approx 0$. i.e., by shifting the reward with proper positive value $b^+$, we are able to initialize the $Q$-value network that lower-bounds the optimal $Q$-value.

choosing an appropriate learning rate is empirically important. We focus on revealing the importance of selecting the universal bias term $b$ in the reward function through the lens of initialization priors in function approximation and show such a bias is generally helpful for both online and offline settings. In the online settings, it improves learning efficiency, and in the offline settings, it promotes conservative exploitation.

# 4   Shifted Priors for $Q$-value Estimation

## 4.1   Reward Shift Equals to Different Initialization

We start by formally introducing notions and the key idea of this work: reward shifting equals different initialization. We use Figure 2 to illustrate how reward shifting affects function approximation, hence changing the learning dynamics for value-based algorithms. The original optimal $Q$-value function is denoted as $Q^*$, and plotted in the figures as **Black** curves. We then denote the shifted optimal $Q$-value function as $Q_{b^+}^*$, which is the $Q$-value function with the shifted reward $r' = r + b^+$. We use **Red** curves in figures to denote those shifted $Q$-value functions. In this section we provide analysis with a positive bias $b^+ > 0$ to illustrate how positively shifted value function affects function approximation, and coordinates the normally applied near-zero initialized function approximators (e.g., neural networks. **Orange** curves in the figure) to inspire conservative behaviors. The discussion of negative biases is elaborated in the next section.

To summarize, shifting the reward function with a positive constant is equivalent to initializing the value network with a smaller value — as the $Q$-value of unseen state-action pairs during training are much lower than their shifted optimal values, those actions will not be selected in argmax-style policy updates — leading to conservative learning behaviors that benefit policy learning in offline settings.

## 4.2   (S1) Offline RL: Conservative Exploitation

According to the key insight we presented in Section 4.1, conservative $Q$-value estimation can emerge with a positively shifted reward. And such a value estimation empirically lower-bounds the optimal $Q$-value function. As shown in Figure 2(a), a positive constant $b^+$ added to the reward function will lead to a universal positively shifted optimal $Q$-value function, and the gap between the primal $Q$-value function and the new one is $\frac{b^+}{1-\gamma}$. Optimizing the $Q$-value approximator with logged data will minimize the difference between the predicted value and the optimal value with observed data. For

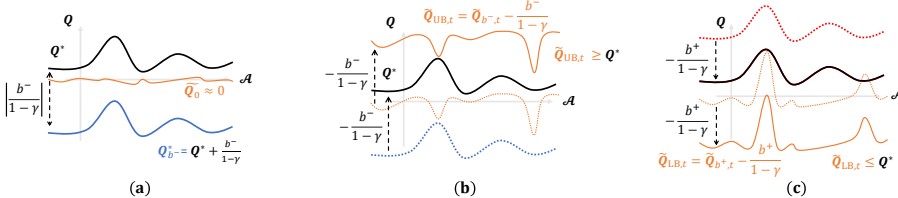

Figure 3: Illustrative figure for curiosity-driven exploration with a negative shifted reward. **(a)** While adding a negative constant value $b^-$ on the reward function leads to negatively shifted optimal $Q$-value function $Q^*_{b^-}$. **(b)** Minimizing the difference between a $Q$-value approximator and the optimal $Q$-value will enable calculating an upper-bound estimation for $Q^*$ which can be used for **optimistic exploration**. **(c)** A positive constant shift added to the reward function can be used for **conservative policy update**.

the unobserved data, the pessimistically initialized $Q$-value approximator guarantees the prediction is lower than the positively shifted optimal ones, thus conservative exploitation can be achieved with argmax-style value propagation and action execution on such a value function.

### 4.3 (Curiosity-Driven) Optimistic Exploration

On the other hand, if we shift the reward function to the negative side, it is equivalent to an optimistic initialization. Figure 3 (a-b) illustrate how adding a negative bias leads to curiosity-driven exploration. With sufficiently small $b^-$ (so that $|b^-|$ larger than the maximal value of any $(s, a)$-pair), such an upper bound of $Q^*$ can be used to conduct curiosity-driven exploration. Intuitively, initializing a value network that always predicts a value larger than the true optimal value will lead to curiosity-driven exploration behavior, as any visited state will be assigned a relatively smaller value and the argmax-style policy will tend to choose under-explored actions.

Based on the analysis above that (1) a positive constant shift added to the reward function can be used for conservative policy updates and (2) a negative constant shift added to the reward function can be used for curiosity-driven exploration, we are ready to access both the upper bound, i.e., the optimistic estimation with $b^-$, and the lower bound, i.e., the conservative estimation with $b^+$ of the optimal value function. Formally, we use

$$\tilde{Q}_{\mathrm{LB},t}(s,a) = \tilde{Q}_{b^+,t}(s,a) - \frac{b^+}{1-\gamma} \qquad (5)$$

to denote the empirical lower bound of value estimation (cf. Figure 2(d), **Orange** line), and

$$\tilde{Q}_{\mathrm{UB},t}(s,a) = \tilde{Q}_{b^-,t}(s,a) - \frac{b^-}{1-\gamma} \qquad (6)$$

to denote the empirical upper bound of value estimation (cf. Figure 3(b), **Orange** line). In both notions, $t$ denotes the optimization step. When $t = 0$, with a near-zero initialization $\tilde{Q}_0 \approx 0$, $\tilde{Q}_{\mathrm{LB},0}(s,a) = -\frac{b^+}{1-\gamma}$ is able to lower bound the unknown optimal $Q$-value given sufficiently large $b^+$. (cf. Figure 2(d), **Green** line). Similarly, $Q$-value is upper-bounded by $\tilde{Q}_{\mathrm{UB},0}(s,a)$ with a sufficiently small $b^-$.

Following those notions, we introduce our sample-efficient algorithms for both continuous control and discrete action space respectively. We propose a practical algorithm for sample-efficient continuous control in Section 4.3.1, and focus on a special class of curiosity-driven exploration method, the RND [15], in Section 4.3.2.

### 4.3.1 (S2) Online RL: Trading-off Exploration and Exploitation with Reward Shift

According to the principle of optimism in the face of uncertainty (OFU), an exploration bonus that manifests the uncertainty of the $Q$-value function can be introduced into the value estimation:

integrating optimistic exploration with conservative exploitation,

$$\hat{Q}(s,a) = \tilde{Q}_{\text{LB},t}(s,a) + \beta[\tilde{Q}_{\text{UB},t}(s,a) - \tilde{Q}_{\text{LB},t}(s,a)]$$

$$= (1-\beta)(\tilde{Q}_{b^+,t}(s,a) - \frac{b^+}{1-\gamma}) + \beta(\tilde{Q}_{b^-,t}(s,a) - \frac{b^-}{1-\gamma}) \tag{7}$$

$$= (1-\beta)\tilde{Q}_{b^+,t}(s,a) + \beta\tilde{Q}_{b^-,t}(s,a) - \frac{(1-\beta)b^+ + \beta b^-}{1-\gamma},$$

where the second term with coefficient $\beta$ denotes exploration bonus that is composed of uncertainty.

For those under-explored state-action pairs, i.e., extremely out-of-distribution samples for our neural network, both $\tilde{Q}_{b^+,t}(s,a)$ and $\tilde{Q}_{b^-,t}(s,a)$ will give near-zero predictions as a consequence of initialization (detailed implications are provided in Appendix C). Hence, the explorative bonus becomes $-\frac{(1-\beta)b^+ + \beta b^-}{1-\gamma}$, which is equivalent to applying another constant reward shift with value of $c_r = (1-\beta)b^+ + \beta b^-$, formally, we have

**Proposition 4.1.** *Assuming we have access to an unbiased estimator for the optimal value function $Q^*$, e.g., with Monte-Carlo estimation $\hat{Q}^* = \mathbb{E}\sum_t \gamma^t r$, and the optimization is based on minimizing the MSE between the unbiased estimator and the function approximator, i.e., $\epsilon_t^2 = (\tilde{Q}_t - \hat{Q}^*)^2$, $\tilde{Q}_t = \tilde{Q}_{t-1} - 2\eta(\tilde{Q}_{t-1} - \hat{Q}^*)$, then the linear combination in Equation (7) is equivalent to a linear combination of the constants with value of $c_r = (1-\beta)b^+ + \beta b^-$.*

The proof can be found in the Appendix B. According to Proposition 4.1, a grid search for trading-off between the three hyper-parameters, i.e., the exploration bias $b^-$, the exploitation bias $b^+$ and the coefficient $\beta$ in Equation (7) is trivial as they only lead to a linear combination as $c_r = (1-\beta)b^+ + \beta b^-$, indicating that

**Corollary 4.2.** *Changing the reward shifting constant $b$ is sufficient to trade off between exploration and exploitation.*

The corollary says, in principle, a meta-learner can be trained to monitor the learning process and select a proper reward shifting constant automatically [54, 55] to balance exploration and exploitation. For the ease of exposition, in this work we choose to focus on the simplest yet effective uniform sampling strategy from multiple shift constants, which has been shown as a strong baseline of those meta-learner approaches [56, 57], and leave more complicated meta-learner-based shifting constant adjustment for future investigation.

Specifically, we use multiple Q-networks to learn with transition tuples $(s, a, r, s')$ sampled from the identical buffer that collects the policy's historcal interactions with the environment. In propagating Q-values through the temporal difference loss, the primal recorded reward value $r$ is replaced with shifted rewards with *different* constant biases to update their individual Q-networks. We then uniformly sample one of those learned Q-networks for the optimization of policy networks (e.g., with DPG [22]). It is worth noting that our approach only requires post-hoc revision of the primal reward function, rather than interacting with the environment multiple times to collect samples for each value network. We dub the proposed method Random Reward Shift (RRS), and provide the pseudo-code in Algorithm 1 of Appendix D.2.

### 4.3.2  (S3) Compatible Curiosity: Tailored Curiosity-Driven Exploration for Value-Based RL

In previous works, the curiosity-driven exploration methods are always work with policy-based methods. In this section, we cast the key insight introduced above to RND [15] — one of the leading algorithms for exploration — to answer why is its vanilla design not suitable for value-based algorithms like Q-learning, and propose a variant of RND that is tailored for DQN.

The vanilla RND use two randomly initialized networks $\phi_1$ and $\phi_2$ to generate the intrinsic reward $r_{\text{int}} = |\phi_1(s,a) - \phi_2(s,a)| \geq 0$ for exploration. During learning, $\phi_2$ is a fixed network and the parameters of $\phi_1$ is optimized to approximate the output of $\phi_2$ for frequently-visited states. We follow Burda et al. [15] to bound the intrinsic reward below 1 and use $r_{\text{int},t}$ to denote the intrinsic reward after $t$ step of optimization. Specifically, $r_{\text{int},0}(s,a) = 1, \forall(s,a)$ — an universally positive bonus is added to the primal reward function at beginning.

According to our analysis in previous sections, such a positive reward shift is equivalent to pessimistic initialization and will lead to conservative behaviors in $Q$-value estimation. Therefore, the exploration

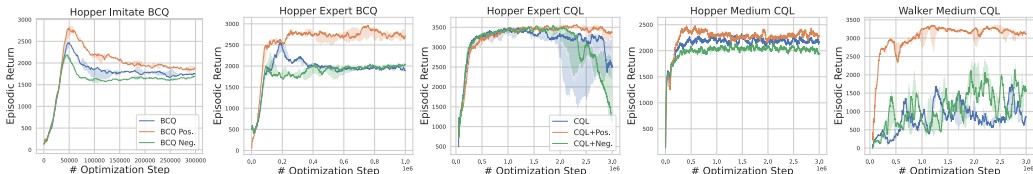

Figure 4: Results on offline RL settings. We verify our key insight that a positive reward shift equals to conservative exploitation thus helps offline value estimation, while a negative reward shift leads to worse performance. Results are from 10 runs with shaded areas indicating the 25%-75% quantiles.

behaviors at the beginning of learning will be hindered, rather than boosted. To overcome such a conservative behavior induced by the pessimistic initialization, we proposed to use $r_{\text{int}}^-(s,a) = |\phi_1(s,a) - \phi_2(s,a)|^2 - I, \forall s,a$, where $I$ is a positive constant that assures $r_{\text{int}}^- \leq 0$ is negatively initialized for optimistic exploratory behaviors.

**A New Type of Curiosity**    To further understand the difference between the curiosity-driven method in ways of intrinsic reward and our reward shifting-based optimistic initialization, let us consider when do those curiosities work in each case: for intrinsic reward methods, an agent must **first experience** a new $(s,a)$ pair before receiving a high intrinsic reward — this is extremely hard with an arg-max style policy. On the other hand, with optimistic initialization, the rarely-visited $(s,a)$ pairs will naturally have higher $Q$-values **before experiencing** it — as the frequently-visited pairs have updated their values with a negatively shifted reward. From such a perspective, reward shifting not only works by itself motivates exploratory behaviors, but can also be seamlessly plugged into intrinsic reward methods to **encourage the first visitation** of new states.

## 5   Experiments

### 5.1   **(S1): Offline RL with Conservative $Q$-value Estimation**

**Experiment Setup**    We start our experiments by demonstrating the effectiveness of reward shifting in offline RL benchmarks. As discussed in Section 4.2, a positive reward shift is equivalent to pessimistic initialization that benefits conservative exploitation. In general, our proposed method can be plugged in to any off-the-shelf offline RL algorithm, we choose to verify the effectiveness and generality of such a conservative $Q$-value estimation based on BCQ [23] and CQL [24], i.e., both distribution-matching approach and conservative value estimation approaches in offline RL. To verify our insight, we experiment with both positive reward shift (**Pos.**) and negative reward shift (**Neg.**), added on either BCQ or CQL.

**Results**    Figure 4 shows our experiment results. We experiment with both the dataset generated in [23] (Hopper Imitate) and the dataset used in [47] (others), and find in our experiments that learning with the CQL dataset is much more stable. The first two panels show results with BCQ as the backbone algorithm. We observe a clear performance drop during the training of BCQ as the policy overfits the batched dataset. Differently, positive reward shift can alleviate such a problem and outperform BCQ in terms of both best-achieved performance and performance after convergence. The following three panels in Figure 4 use CQL as the backbone. Implementation details and more results can be found in Appendix D.1.

**Take-Away Message**    In all experiments, shifting the reward with a positive constant improves learning performance, while a negative reward shift impedes to efficient learning — as expected.

### 5.2   **(S2): Online RL with Randomized Priors**

**Experiment Setup**    We then conduct experiments in the MuJoCo locomotion benchmarks to demonstrate reward shifting improves learning efficiency in the online RL settings. As our implementation is based on TD3, we use TD3-based variants as our baselines: The **TD3** is trained with default settings according to  Fujimoto et al. [14]. We also include **Ensemble TD3** and **Bootstrapped TD3** as baselines due to they are similar to our work in using multiple $Q$-networks in value estimation.

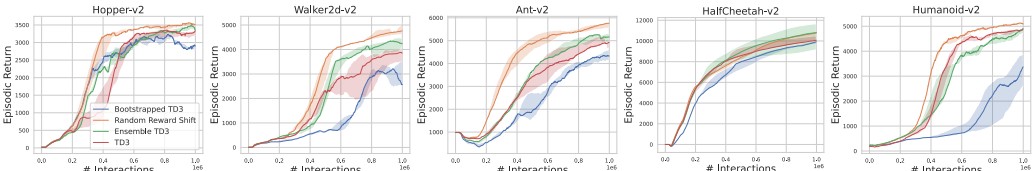

Figure 5: Results on continuous control tasks, Random Reward Shift (RRS) outperforms its value-based baselines in most environments. Results are from 10 runs with shaded areas indicating the 25%-75% quantiles.

We follow Osband et al. [45] but extend it to the continuous control settings. In continuous control settings, the argmax operator is approximated by the policy network, and multiple policy networks are needed to cooperate with the multiple bootstrapped $Q$-value networks. Otherwise, multiple $Q$-value networks are not independent of each other thus breaking the condition of bootstrapped value estimation. The Ensemble TD3 presents the baseline performance when multiple $Q$-networks are used for value estimation in TD3, and works as an ablation of our method where all shift priors are set to be $0$.

As has been illustrated in Sec. 4.3.1, learning with different reward shifting values is equivalent to learning with optimistic or conservative initialization. In our instantiating of RRS, we use $3$ $Q$-networks with different priors. We empirically find $\pm 0.5, 0$ work universally good for all environments. Though, future investigation on hyper-parameter may help to further improve the performance.

**Results** Results are shown in Figure 5. RRS outperforms the vanilla TD3 in all five environments and outperforms all baseline methods in most tasks. In the experiment of Bootstrapped TD3 and Ensemble TD3, we also use $3$ $Q$-networks for a fair comparison. Note that there is a trade-off between computational complexity and sample efficiency, i.e., using more $Q$-networks may further improve the performance at the cost of more computational expenses, as reported in [45]. More implementation details, pseudo-code of RRS, and ablation studies can be found in Appendix D.2.

**Take-Away Message** Shifting the reward function can trade-off between exploration and exploitation. The ensemble performance of multiple value networks with random reward shifting drastically improve learning efficiency in continuous control.

### 5.3 (S3): Optimistic Random Network Distillation

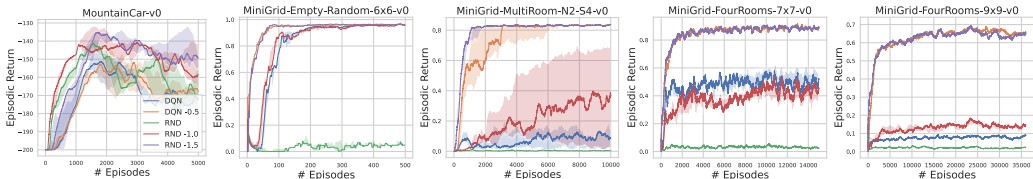

Figure 6: Value-based RND with shifted prior: Plugging the vanilla RND into DQN is not well-motivated according to our analysis in Section 4.3.2. The insight of equivalence between negative reward shifting and curiosity-driven exploration motivates us to negatively shift the intrinsic reward of RND, which drastically improves DQN-based RND. Results are from 10 runs with shaded areas indicating the 25%-75% quantiles.

**Experiment Setting** To demonstrate the key insight and effectiveness of reward shifting in optimistic exploration, we benchmark with five discrete exploration tasks, including the classic MountainCar control and four MiniGrid navigation tasks [58], environment details can be found at Appendix D.3.

**Results** Figure 6 presents the results of 5 different methods for comparison: besides the as-is ①**DQN** and ②**RND** baselines, we use ③**DQN -0.5** to indicate DQN with a $-0.5$ reward shift, and ④**RND -1.0** ⑤ **RND -1.5** to indicate RND with $-1.0, -1.5$ reward shift, respectively. In the following, we use $\succ$ between methods to indicate the former outperforms the latter. Comparing the results:

**1. reward shifting is equivalent to optimistic initialization, thus boosting exploration**
③ ≻ ①: a negative reward shift is equivalent to optimistic initialization and helps exploration.

**2. RND is effective for value-based exploration — as long as a negative intrinsic reward is used**
① ≻ ②: vanilla RND with positive intrinsic reward is always worse than DQN: adding a positive intrinsic reward like RND is harmful for exploration as it is equivalent to a pessimistic initialization. ④ ≻①; ⑤ ≻ ③: RND is effective for exploration (i.e., improve over DQN) as long as the intrinsic reward bonus is always negative.

**3. reward shifting for optimistic initialization is orthogonal to other exploration methods**
⑤ ≻ ④: increasing the magnitude of the negatively shifted reward can further improve the exploration performance — reward shifting can either work in isolation or get combined with other exploration algorithms as they are working orthogonally. [4]

**Take-Away Message**   Reward shifting is equivalent to optimistic initialization, it can help with exploration in value-based methods. Importantly, such an intrinsic motivation is orthogonal to previous count-based methods, such that it can either work in isolation or be combined with conventional curiosity-driven exploration methods.

# 6   Conclusion

In this work, we studied how reward shifting affects policy learning in value-based deep reinforcement learning algorithms. Although constant reward shift should not change the optimal policy induced by the optimal value function, in practice such a constant shift *does affect the function approximation*, and leads to different learning behaviors. Our detailed analysis manifests the fact that a constant reward shift is equivalent to using different initialization in the value function approximation. The proposed idea is then verified through a variety of application settings. Specifically, we show that a negative reward shift leads to curiosity-driven exploration, while a positive reward shift helps conservative exploitation. Importantly, our analysis reveals that changing reward shifting constant itself is sufficient in trading-off between exploration and exploitation. We empirically verify the effectiveness of the proposed method in a variety of experiments, including better exploitation in offline RL, sample-efficient learning in continuous control benchmarks, and enhanced curiosity-driven exploration in value-based discrete control.

While our experiments demonstrate the performance gain is quite robust to the shifting constant, we would like to point out that the theoretical guidance for such a shifting constant is missing. Potential solutions may lie in analysis from the perspective of optimization for black-box models, yet it is out of the scope of the current empirical study and left for future research.

# 7   Acknowledgement

We thank all anonymous reviewers, ACs, PCs for their efforts and time in the reviewing process and in improving our paper. This work is done with the warm supports from the MMLab members. We acknowledge the insightful discussions with Ziping Xu, the Hot Spring Harbor group and the van der Schaar Lab members in improving the presentations of this paper. We thank Takuya Kanazawa in pointing out the concurrent work of Dubey et al. [59] that also discusses the effects of shifting terms in reward function. We thank all insightful comments and discussions during the conference poster sessions that helps us in improving the presentation of the paper.

---

[4]Code is available at GitHub

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
