# A Extended Related Work

We extend our related work section on the following related topics as suggested by reviewers:

## A.1 Discussion on Ensembles and Distributional RL

In Distributional RL literature [60–64], the distribution of the $Q$-value, rather than the mean scaler, is estimated. Distributional-RL focuses on stochastic reward mechanisms and smooth the temporal difference learning in a distributional level, and can be applied to risk-sensitive scenarios [65] where the worst-case performance can be controlled [66]. In those scenarios, the systematic uncertainty is the crucial issue to address, whereas in our work, we focus on deterministic transition dynamics and use OFU to tackle the epistemic uncertainty in the section of RRS.

Several previous works discussed ensemble methods for exploration, both for discrete control [67, 68] and for continuous control [69, 70]. However, in our work we show that more exploratory behavior can emerge with the help of reward shifting under only single $Q$-value network — as a proof of concept that negative reward shift is equivalent to optimistic initialization.

## A.2 Model-Based Exploration and Uncertainty Estimation

Besides the model-free value-based RL methods we focused in our paper, there exist literature like R-Max [40] that work with model-based methods for better exploration. Moreover, combining multiple neural networks for uncertainty estimation is well-established in supervised learning [71], and similari idea has been explored in the context of RL for discrete control [45, 72]. In this work, we showcase that a diversified set of reward shifting constants can work as priors for such an ensemble.

Given the ensemble networks without ground-truth, it is in general hard to disentangle the aleatoric uncertainty from epistemic uncertainty. Our method works on deterministic reward and transition dynamics to circumvent the discussion of uncertainty stratification. In the deterministic settings, the source of uncertainty can be solely attributed to the epistemic uncertainty and hence help informative exploration. When it comes to the stochastic environments, the entanglement of aleatoric uncertainty and epistemic uncertainty will make the problem much more difficult [73] as intrinsic motivation methods may get trapped by pursuing the actions that result in high aleatoric uncertainty in certain circumstances (e.g., the Noisy-TV) [34].

# B Proof of Proposition 4.1

*Proof.* the estimated $Q$-value $\hat{Q}(s,a)$ is composed by the two estimators with function approximation error, defined as $\epsilon_{b^+}(s,a) = \tilde{Q}_{b^+,t}(s,a) - \frac{b^+}{1-\gamma} - Q^*(s,a)$, and $\epsilon_{b^-}(s,a) = \tilde{Q}_{b^-,t}(s,a) - \frac{b^-}{1-\gamma} - Q^*(s,a)$.

$$
\begin{aligned}
&(1-\beta)\epsilon_{A,t} + \beta\epsilon_{B,t} \\
&= 2\eta(1-\beta)\hat{Q}^* + (1-\beta)(1-2\eta)\tilde{Q}_{A,t} + 2\eta\beta\hat{Q}^* + \beta(1-2\eta)\tilde{Q}_{B,t} \\
&= 2\eta\hat{Q}^* + (1-2\eta)[(1-\beta)\tilde{Q}_{A,t} + \beta\tilde{Q}_{B,t}] \\
&= 2\eta\hat{Q}^* + (1-2\eta)[(1-\beta)(1-2\eta)^t\tilde{Q}_{A,0} + \frac{4(1-\beta)\eta^2}{1-(1-2\eta)^t}\hat{Q}^* + \beta(1-2\eta)^t\tilde{Q}_{B,0} + \frac{4\beta\eta^2}{1-(1-2\eta)^t}\hat{Q}^*] \\
&= 2\eta\hat{Q}^* + (1-2\eta)[(1-2\eta)^t((1-\beta)\tilde{Q}_{A,0} + \beta\tilde{Q}_{B,0}) + \frac{4\eta^2}{1-(1-2\eta)^t}\hat{Q}^*] \\
&= \epsilon_{C,t}
\end{aligned}
\tag{8}
$$

where $C = (1-\beta)A + \beta B$ and the last line requires $\tilde{Q}_{A,0} = \tilde{Q}_{B,0} = \tilde{Q}_{C,0}$ are identical initialization.

With this notion, Equation (7) can be re-written as

$$\hat{Q}(s,a) = Q^*(s,a) + (1-\beta)\epsilon_{b^+}(s,a) + \beta\epsilon_{b^-}(s,a)$$
$$= Q^*(s,a) + \epsilon_{(1-\beta)b^+ + \beta b^-}(s,a) \qquad (9)$$
$$= Q^*(s,a) + \epsilon_{c_r}(s,a)$$

where the second line relies on the linear assumption of the approximation error $(1-\beta)\epsilon_{b^+}(s,a) + \beta\epsilon_{b^-}(s,a)$. We further have $(1-\beta)\tilde{Q}_{b^+} + \beta\tilde{Q}_{b^-} = \tilde{Q}_{(1-\beta)b^+ + \beta b^-}$ and $\hat{Q}(s,a) = \tilde{Q}_{c_r}(s,a)$, telling us that trading-off between the constant $b^-$ used for exploration and the constant $b^+$ used for exploitation with the coefficient $\beta$ is equivalent to use another constant with value of $c_r = (1-\beta)b^+ + \beta b^-$.

$\square$

## C   Implications of Assumption in Section 4.3.1

In our main text, the estimated values for extremely o.o.d. samples are assumed to be near zeros. We provide detailed implications and explanations in this section.

On the one hand, it's clear that such an assumption holds for the tabular settings, that un-visited state-action pairs have the value in tabular initialization.

On the other hand, we acknowledge it as a mild assumption that there always exists o.o.d. samples that have the $Q$-values near zero for function approximation settings. Interpolation between those o.o.d. samples and other state-action pairs will clearly lead to an "in-between" value estimation, which in practice can be achieved with properly regularized neural networks.

The key insight we want to emphasize in Section 4.3.1 is that for frequently visited state-action pairs, the value discrepancy with different initialization are small, while for seldomly-visited state-action pairs, the discrepancy are relatively large, enabling the usage of such discrepancy as exploration bonus.

## D   Implementation Details and Ablation Studies

**Hardware and Training Time**   We experiment on a server with 8 TITAN X GPUs and 32 Intel(R) E5-2640 CPUs. In general, shifting the reward does not introduce further computation burden except in the continuous control tasks, our method of Random Reward Shift (RRS) requires two additional $Q$-value networks. In our PyTorch-based implementation, those additional networks can be easily implemented and optimized in a parallel manner, and the extra computational burden is equivalent to using a $\sqrt{3}$ times wider neural network during optimization. It is worth noting that RRS is computationally much cheaper than the Bootstrapped TD3, where additional policy networks are also needed.

**Network Structure**   Our implementation of TD3, BCQ and CQL are based on code released by the authors, without changing hyper-parameters. We implement DQN based on a 3-layer fully connected neural network with 64 hidden units for the $Q$-value function, using ReLU and linear activation respectively. We use the Adam optimizer with learning rate of 0.001, and use an epsilon-greedy approach as naive exploration strategy. In our RND, we use two 4-layer fully connected neural networks with 512 units and ReLU activation in each hidden layer, and a softmax activation for the output layer. Adam optimizer is used for the optimization of the RND networks with learning rate 0.0001.

Our code is provided in the supplementary materials, and will be made public available.

### D.1   Offline RL

In our experiments, we use a fixed dataset with 10k offline trainsition tuples for offline RL learning. Our implementation of BCQ and CQL are both based on the code provided by the authors. The only change we made to verify our insight is to shift the reward by a constant. In most environments, we

**Algorithm 1** Sample-Efficient Continuous Control with Random Reward Shift

**Require**
  Size of mini-batch $N$, smoothing factor $\tau > 0$, $K$ reward shift values $r'_k = r + b_k, k = 1, \ldots, K$.
  Random initialized policy network $\pi_\theta$, target policy network $\pi_{\theta'}$, $\theta' \leftarrow \theta$.
  $K$ random initialized $Q$ networks, and corresponding target networks, parameterized by $w_k, w'_k, w'_k \leftarrow w_k$ for $k = 1, \ldots, K$. (e.g., a ModuleList in PyTorch).
**for** iteration $= 1, 2, \ldots$ **do**
  Uniformly sample one of the $K$ $Q$-functions, $Q_{w_k}$, for policy update
  **for** t $= 1, 2, \ldots$ **do**
    # Interaction
    Run policy $\pi_\theta$, and collect transition tuples $(s_t, a_t, s'_t, r_t)$.
    Sample a mini-batch of transition tuples $\{(s, a, s', r)_i\}_{i=1}^N$.
    # Update $Q_w$ (in parallel)
    Calculate the $k$-th target $Q$ value $y_{k,i} = r_i + b_k + Q_{w'_k}(s'_i, \pi_{\theta'}(s'_i))$
    Update $w_k$ with loss $\sum_{i=1}^N (y_{k,i} - Q_{w_k}(s_i, a_i))^2$.
    # Update $\pi_\theta$
    Update policy $\pi_\theta$ with $Q_{w_k}$
  **end for**
  # Update target networks
  $\theta' \leftarrow \tau\theta + (1 - \tau)\theta'$.
  $w'_k \leftarrow \tau w_k + (1 - \tau)w'_k, k = 1, \ldots, K$.
**end for**

find $r' = r + 8$ provides good enough performance. While in Hopper Medium CQL we find using a smaller positive reward shift $r' = r + 1$ works better than $r' = r + 8$, and for Walker Medium CQL, using a larger reward shift of $r' = r + 50$ further improves the result with $r' = r + 8$.

Figure 7 shows different performance under different choices of the reward shift constant. We denote a positive reward shift $r' = r + 8$ as **Pos.1**, denote $r' = r + 20$ as **Pos.2** and denote $r' = r + 50$ as **Pos.3** for all experiments excetp in the Hopper Medium CQL we use **Pos.1** to denote $r' = r + 1$.

In the experiments based on BCQ (first two figures). We can observe a uniformly performance improvement with all choices of reward shift constants. As the algorithm of CQL has already taken the conservative value estimation into consideration, in the experiments based on CQL, the performance is more closely related to the constant we use. Specifically, in Hopper Expert, while using any of the positive reward shift constants improve the learning stability, $r' = r + 8$ performs better on preserving the learning efficiency during early learning stage. For Hopper Medium, we find using larger positive constants hinder the performance. For Walker Medium, using a larger constant in reward shift performs much better than using a smaller one.

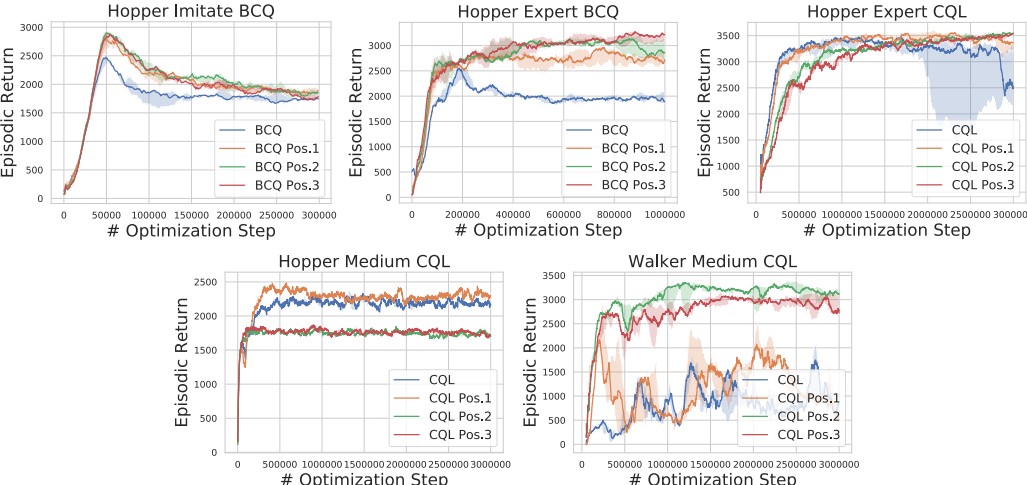

Figure 7: Performance with different reward shift constants.

## D.2   Continuous Control

**Details of RRS**   Although we find in the motivating example that a $-5$ reward shift is able to remarkably improve the asymptotic performance of TD3, in this work we aim at proposing an uniformly suitable method based on the insight behind the motivating example. Therefore we propose to use $\pm 0.5, 0$ as the reward shifting constants. We find in experiment that the sampling frequency does not affect the performance. And in the experiments we follow BDQN [45] to use a fixed value network throughout a whole trajectory. i.e., one of the $K$ $Q$-networks is sampled uniformly after each episode with length of 1000 timesteps. Intuitively, searching for more suitable reward randomization designs may further improve the performance, yet that is beyond the coverage of this work.

**Ablation Studies**   We experiment with different number of $Q$-value networks as well as different choices of the random reward shifting ranges. Results are presented in Figure 8. We denote RRS with 7 reward shifting constants ( and therefore also 7 $Q$-networks) as **RRS-7**, and denote RRS with 3 reward shifting constants ( and therefore also 3 $Q$-networks) as **RRS-3**. The constants following **RRS-3/RRS-7** are the ranges of those random constants. Specifically, we use $[-0.5, 0, 0.5]$ for the **RRS-3 0.5** settings, $[-1.0, 0, 1.0]$ for the **RRS-3 1.0** settings, $[-0.5, -0.33, -0.17, 0, 0.17, 0.33, 0.5]$ for the **RRS-7 0.5** settings and $[-1.0, -0.67, -0.33, 0, 0.33, 0.67, 1.0]$ for the **RRS-7 1.0** settings. According to the experimental results, RRS is not sensitive to hyper-parameters, showing the robustness of the proposed method. We believe further search for those hyper-parameters can further improve the learning efficiency, yet this is off the main scope of this work and therefore left for the future research.

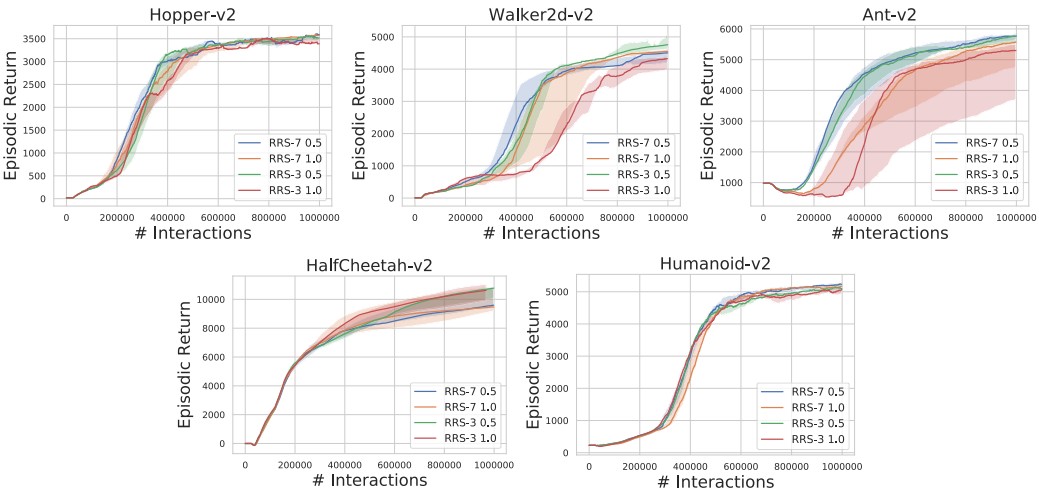

Figure 8: Performance with different reward shift constants and different number of $Q$-networks.

## D.3   Random Network Distillation

**Environments**   In this work, we experiment with five discrete (sparse reward) exploration tasks , namely the MountainCar-v0, and four navigation tasks of MiniGrid suite [58], namely the task of Empty-Random, MultiRoom, and FourRooms, to verify our insight on improving RND for value-based curiosity-driven exploration. Figure 9 shows example of different tasks.

**Hyper-Parameter Settings**   We use a 2-layer NN with 64 hidden units for $Q$-networks in DQN and set RND networks to be 3-layer NN with 512 hidden units. $\epsilon$-greedy exploration is applied to DQN with $\epsilon$ decays from 0.9 to 0.05 in the first 1/5 episodes. Size of replaybuffer is set to be 100000.

**Ablation Studies**   We experiment with different reward shifting constants in the discrete control settings.     We use a relatively large range in choosing constants, i.e., $\{-0.05, -0.15, -1.0, -1.5, -2.0, -2.5, -5.0, -10.0\}$. Results are presented in Figure 10. In all experiments, using a moderate reward shifting constant like $\{-1.0, -1.5, -2.0, -2.5\}$ remarkably improves the learning efficiency. On the other hand, a too aggressive reward shifting will lead to too much curiosity exploration and hinder the learning efficiency in the limited number of interactions.

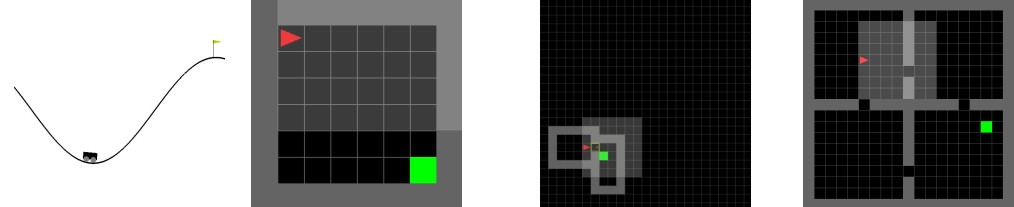

Figure 9: Examples of environments used in Section 5.3. The first figure shows the MountainCar-v0 environment where a car needs to accumulate potential energy to reach the flag, to receive a positive reward. The second figure shows the maze of the Empty-Random task with size of 6, the third one shows the MultiRoom of level S2-N4, where there are 2 rooms with size 4, the last figure shows example of FourRoom task with size 17. In our experiments, as we use the vanilla DQN as the baseline, which is not suitable for partial observable tasks, we use a smaller maze of size 7 and 9 to avoid further dependency on memories. In all tasks of the MiniGrid domain, the triangular red agent need to navigate to the green goal square, and the observable region is only a 7x7 square the agent is facing to (i.e., the regions with shallower color in the last three figures).

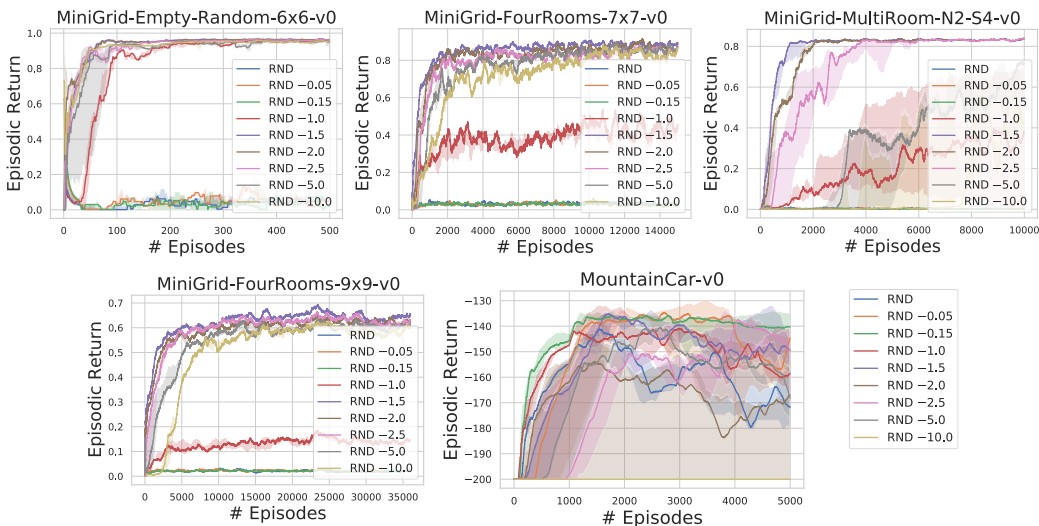

Figure 10: Performance with different reward shift constants in RND.

# E   Additional Experiments

## E.1   Necessity of Explorative Behaviors in Maze Tasks

We demonstrate the benefits of reward shifting for deep-exploration tasks and the on-par performance of reward shifting for easy tasks that does not require deep-exploration. We experiment on the maze environment and change the size of maze to vary from 2 to 20, denoting as S2, S5, S10, S15, S20, separately. Results averaged over 8 runs are reported in Figure 11. We find in easy tasks (S2, S5, S10), both count-based exploration and reward shifting perform similarly to the $\epsilon$-greedy exploration, while on challenging tasks (S15, S20), more explorative behavior encouraged by reward shifting and count-based exploration are important for efficient learning.

## E.2   Performance in Challenging Continuous Control Exploration Tasks

Figure 12 show experiments on (1) the HalfCheetah-SparseReward, where a reward of +1 is provided only when the forward movement of the halfcheetah is larger than 5 unit with regard to the timeframe; We note that this environment is different from the SparseHalfCheetah environment that first introduced in VIME [74], as we use different time frames. (We acknowledge and thank the anonymous reviewer FqMf for pointing out this difference.) and (2) the Humanoid, which is a high dimensional continuous control task in the MuJoCo locomotion suite; to verify the effectiveness of reward shifting in exploration. In the HalfCheetah-SparseReward environment, the maximum score is 1000 for each

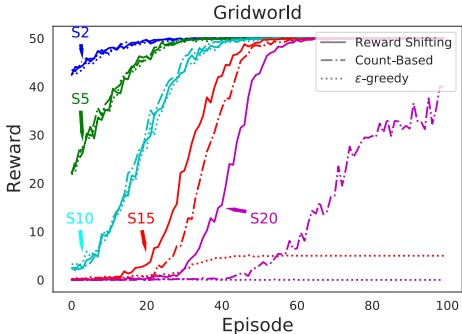

Figure 11: In deep-exploration tasks, reward shifting benefits exploration by optimistic initialization, while in easier tasks, reward shifting does not hinder exploitation, convergence efficiency, and the asymptotic performance.

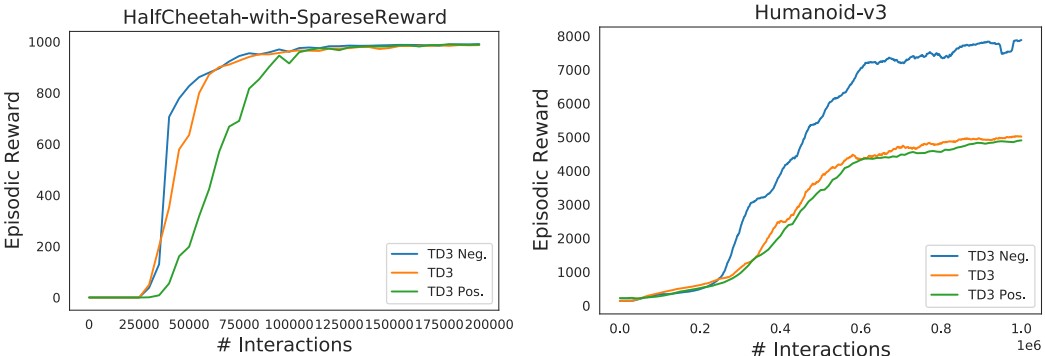

Figure 12: Experiments on two challenging continuous control tasks. Experiments are repeated with 8 runs.

episode. We use $-0.5$ as the negative reward shift for exploration and use $0.5$ for comparison. In Humanoid, the per-step reward is approximately $5$ in previous well-performing agents [14, 20], and we hereby use negative shift $-5$ and use positive shift $5$, for comparison.

In HalfCheetah-SparseReward, we find a negative reward shift lead to more explorative behavior and improves the learning efficiency while a positive reward shift hinders the learning efficiency. In Humanoid, we find using a reward shift can drastically improve the asymptotic performance by $+60\%$, while learning with a positive reward shift retard the learning and converge to a lower performance.

## E.3 Performance in Goal-Conditioned Continuous Control (Robotics) Suite

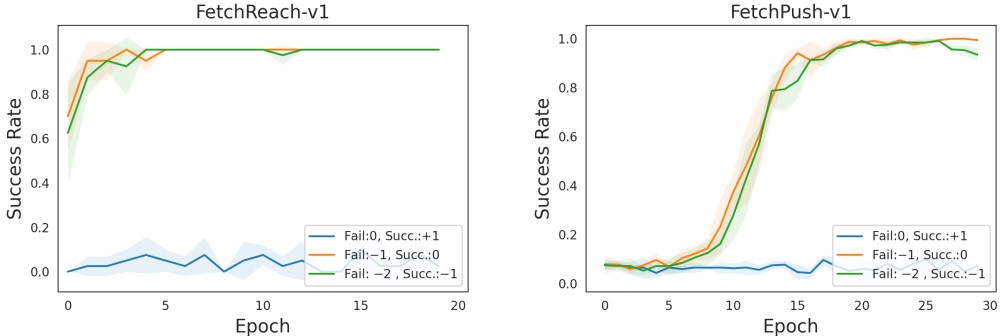

Figure 13: Experiments on two GCRL robotics tasks. Experiments are repeated with $4$ runs.

To further address the reviewer's concern on the applicability of reward shifting on challenging continuous control exploration tasks, we benchmark reward shifting on the FetchRobotics suite [8] that is usurally considered to be challenging exploration task in GCRL literature [75, 76]. Figure 13 shows the results we get on FetchReach-v1 and FetchPush-v1 environment. We use HER [76] as the backbone algorithms and vary the reward for failure and success in achieving the goals. In their default setting, reaching the goal will receive a reward of $0$, otherwise, the agent will receive $-1$ as punishment. In experiments, we find using a positive reward $+1$ in reaching the goals while using a trivial $0$ reward otherwise will drastically hinder the learning efficiency of HER. Similar empirical discovery has been reported in Sun et al. [37] in the PPO-based learners. This set of experiments verifies our key insight one more time that explorative behaviors emerge with a negatively shifted reward function, and a positive reward shift leads to conservative behavior.

To sum up, our key insight reveals the mechanism of how such an empirically verified heuristic design in GCRL works: a negative reward $-1$ (also interpreted as cost) works in the same way as reward shifting to improve exploration.