# OpenReview forum: "Exploit Reward Shifting in Value-Based Deep-RL: Optimistic Curiosity-Based Exploration and Conservative Exploitation via Linear Reward Shaping"
_NeurIPS.cc/2022/Conference — NeurIPS 2022 Accept_

### Official Review · Reviewer_FqMf · 2022-07-08

**Rating:** 6
**Confidence:** 4
**Soundness:** 3 good
**Presentation:** 2 fair
**Contribution:** 3 good

**Summary:**

This paper studies how reward shifting can be used to encourage or discourage exploration without modifying optimal behavior in deep Q-learning algorithms. The authors demonstrate how reward shifting can be used in different contexts where exploration is desirable or undesirable and show that it can improve performance on a range of tasks.

**Questions:**

Some other suggestions (less critical stuff):

-Consider using a non-gridworld example for figure 1. This paper is mostly concerned with deep RL and continuous MDPs, so leading with gridworld results seems odd.

-Figures 2 and 3 I found kind of confusing. I see and like the intent, but it's hard to piece together what's happening based on the caption and equations in the panels. Consider cleaning up the captions and labeling the lines with descriptive labels rather than equations/symbols that haven't been fully introduced/explained at that point in the paper.

-I don't like table 1. It's a matter of preference, but I don't find it informative. Consider focusing on the domains/use-cases for reward shifting instead of the comparison to previous methods.

-I didn't understand the point about RND-1.0 being compared to vanilla DQN. RND has a variable reward bonus, so I'm not sure why it should neatly cancel out when reward shifted.



**Limitations:**

This paper doesn't discuss the limitations of reward shifting, which I think is a significant issue. The experiments presented show reward shifting to work well on a variety of tasks, but when does it perform poorly?

I think it might have issues with stochastic rewards, for example (offhand it seems like it might work but with worse bounds?)

How does reward shifting perform on harder exploration tasks, or larger/more complex off-policy datasets? Even if it under-performs compared to dedicated methods there is value in a simpler and more general but less specialized algorithm, so I'd love to see the results regardless of outcome.

Regarding societal impact, I don't think this work has any direct negative impact risks.

**Strengths And Weaknesses:**

The core idea proposed by this paper, reward shifting, is shown to be both simple and effective. A simple method (both conceptually and to implement) is more likely to see wider adoption and further work, so that is a significant strength of this paper.

I do feel there are a number of significant issues to be addressed to make this paper great, however. To be clear, I like this work, but I find it hard to recommend it be accepted without substantial modifications (particularly the points below). If these issues can be addressed I think this will be a good paper and a valuable contribution to the field.

-The clarify of the writing is poor. I found the paper generally understandable, so this isn't a fatal flaw, but it made understanding more difficult than it needed to be. Apologies if I'm mistaken, but this paper comes across as written by scientists not fluent in English, and regardless of acceptance status I strongly encourage the authors to take a careful editorial pass (preferably with an English-fluent colleague's help) to improve the language. This will greatly improve the presentation and lead to more people reading and citing this paper in the future.

-The paper feels somewhat padded. Given the simplicity of the core idea it doesn't take much space to explain how reward shifting works, and the paper as is (IMO) spends more space than needed on it. This space could be better spent addressing some of the points below, or increasing the size of the results figures (which are a little too small to read without zooming in).

-The idea of reward shifting is not novel, though I do not know of any work that explores it in the way this paper does. I think the novel contribution is significant and worthy of publication here, but a longer discussion of related work seems important for a paper like this one that applies an old idea in a new way to good effect.
 --There is related work using value function initialization, which is equivalent to reward shifting, to perform exploration. For example DORA, which is cited in this paper but not discussed, uses biased initialization for exploration, which should perform similarly to reward shifting, and seems like a natural point of comparison.
--Similarly, biased value ensembles have been proposed before, but such methods don't seem to be discussed (Some related works: "A distributional code for value in dopamine-based reinforcement learning" by Dabney et al, "UCB Exploration via Q-Ensembles" by Chen et al)

-The experiments cover several cases and domains, which is good, but the comparison to RND seems odd as it focuses on simple minigrid environments and doesn't include any hard exploration continuous MDP tasks where baseline naive exploration methods perform poorly, which were the intended use case for RND. For example, the Sparse HalfCheetah environment proposed by VIME or other continuous control exploration tasks similar to the MuJoCo gym tasks already tested would be easy to compare on without additional modification to the algorithm.

-I'd like to see some exploration of the limitations of reward shifting (see limitations section).

---

> ### Author Response · Authors · 2022-08-02
> **Response to Reviewer FqMf [Part 3/3]**
>
>
>
> ---
> ### References
>
> [1] Bellemare, Marc G., Will Dabney, and Rémi Munos. "A distributional perspective on reinforcement learning." International Conference on Machine Learning. PMLR, 2017.
>
> [2] Dabney, Will, et al. "Distributional reinforcement learning with quantile regression." Proceedings of the AAAI Conference on Artificial Intelligence. Vol. 32. No. 1. 2018.
>
> [3] Lyle, Clare, Marc G. Bellemare, and Pablo Samuel Castro. "A comparative analysis of expected and distributional reinforcement learning." Proceedings of the AAAI Conference on Artificial Intelligence. Vol. 33. No. 01. 2019.
>
> [4] Barth-Maron, Gabriel, et al. "Distributed distributional deterministic policy gradients." arXiv preprint arXiv:1804.08617 (2018).
>
> [5]Dabney, Will, et al. "A distributional code for value in dopamine-based reinforcement learning." Nature 577.7792 (2020): 671-675.
>
> [6] Urpí, Núria Armengol, Sebastian Curi, and Andreas Krause. "Risk-averse offline reinforcement learning." arXiv preprint arXiv:2102.05371 (2021).
>
> [7] Delétang, Grégoire, et al. "Model-Free Risk-Sensitive Reinforcement Learning." arXiv preprint arXiv:2111.02907 (2021).
>
> [8] Chen, Richard Y., et al. "Ucb exploration via q-ensembles." arXiv preprint arXiv:1706.01502 (2017).
>
> [9]Chen, Richard Y., et al. "UCB and infogain exploration via q-ensembles." arXiv preprint arXiv:1706.01502 9 (2017).
>
> [10] Lee, Kimin, et al. "Sunrise: A simple unified framework for ensemble learning in deep reinforcement learning." International Conference on Machine Learning. PMLR, 2021.
>
> [11] An, Gaon, et al. "Uncertainty-based offline reinforcement learning with diversified q-ensemble." Advances in neural information processing systems 34 (2021): 7436-7447.
>
> [12] Chen, Xinyue, et al. "Randomized ensembled double q-learning: Learning fast without a model." arXiv preprint arXiv:2101.05982 (2021).
>
> [13] Clements, William R., et al. "Estimating risk and uncertainty in deep reinforcement learning." arXiv preprint arXiv:1905.09638 (2019).
>
> [14] Burda, Yuri, et al. "Large-scale study of curiosity-driven learning."  arXiv preprint arXiv:1808.04355 (2018).

---

> ### Author Response · Authors · 2022-08-02
> **Response to Reviewer FqMf [Part 2/3]**
>
>
>
> ### Q5: Motivating Example in Figure 1.
> - In figure 1, we provided a simple example in a tabular case to show the (negative) reward shifting is a simple yet effective way of boosting exploration. We tried to make the motivating example clear enough for readers to grasp the key insight of our work and develop the insight into different settings later in our work. An alternative choice is to use the continuous control task examples like SparseHalfCheetah or Humanoid, but in this way, there is no clear analogy as Count-Based exploration in the tabular cases and will be relatively more complicated as a motivating example.
>
>
> ### Q6: Table 1
> - We use Table 1 to manifest that reward shifting is a generally applicable method that covers both exploration and exploitation (offline RL). We don’t intend to compare against other methods but aim to contrastively present the flexibility and generality of reward shifting. At the same time, Table 1 contextualizes the novelty of reward shifting with previous works. We’ve updated the caption and part of the layout of Table 1 to make it clearer.
>
>
> ### Q7: Experiment design in RND v.s. DQN
> - (a) To demonstrate the effectiveness of reward shifting in discrete action space exploration, comparing the results of DQN and DQN + reward shifting is enough to draw the conclusion.
>
> - (b) On the other hand, in previous works, RND is always combined with policy-based discrete control algorithms like PPO. In our work, we want to use our key insight to answer the question that why naive attempts of combining value-based methods like DQN with RND will fail — adding a positive intrinsic reward like RND is harmful for exploration as it is equivalent to a pessimistic initialization.
>
> - (c) According to Figure 6, we can find:
>   - (1). DQN > RND: vanilla RND with positive intrinsic reward is always worse than DQN: adding a positive intrinsic reward like RND is harmful for exploration as it is equivalent to a pessimistic initialization.
>   - (2). DQN-0.5 > DQN: adding a negative reward shifting is equivalent to optimistic initialization and helps exploration.
>   - (3). RND-1.0 > DQN; RND-1.5 > DQN-0.5: RND is effective for exploration (i.e., improve over DQN) as long as the intrinsic reward bonus is always negative.
>   - (4). RND-1.5 > RND-1.0: increasing the magnitude of the negatively shifted reward can further improve the exploration performance — reward shifting can either work in isolation or get combined with other exploration algorithms as they are working orthogonally.
>
> - (d) We’ve entirely rephrased the presentation in Section 4.3.2 and Section 5.3 to make it clearer.
>
> ### Q8:  when does it perform poorly?
> - Our experiments have shown that for conservative exploration tasks, reward shifting may perform poorer than the baselines when the shifting constant is too large (e.g., shifting constant = 50, which is approximately 10 - 20 times larger than the maximal step-wise reward of the environment. cf. Figure 7.), as it can be over-pessimistic on extrapolations.
> For optimistic exploration tasks, reward shifting also may perform pooer than the baselines when the shifting constant is too large (e.g., shifting constant = -10, which is more than 10 times larger than the potential episodic return for navigation tasks. cf. Figure 10.).
>
>
> ### Q9: Stochastic Reward
> - In this work, we focused on deterministic reward and transition dynamics. In the deterministic settings, the source of uncertainty can be solely attributed to the epistemic uncertainty and hence help informative exploration. When it comes to stochastic environments, the entanglement of aleatoric uncertainty and epistemic uncertainty will make the problem much more difficult [13] as intrinsic motivation methods may get trapped by pursuing the actions that result in high aleatoric uncertainty in certain circumstances (e.g., the Noisy-TV) [14]. Using reward shifting to disentangle those two sources of uncertainty is a promising future direction.

---

> ### Author Response · Authors · 2022-08-02
> **Response to Reviewer FqMf [Part 1/3]**
>
>
>
> Thank you for your elaborate feedback. We will address each question and weakness in turn.
>
> ---
>
> ### Q1: Language/ Writing.
> - We’ve updated the draft and mainly re-write the following sections to address this problem, including but not limited to:
>   - In the related work section, we include discussions on the works mentioned by the reviewer, and updated the phrasing in Table 1. We added extended discussions on related work at Appendix E.
>   - In the method sections, we updated the presentation and formally introduce the notions before figures. We also updated the captions to make it clearer.
>   - We’ve entirely rephrased the presentation in section 4.3.2 and section 5.3.
>
>
> ### Q2: Extended Related Work
> - (a) We’ve updated our related work section and extended the discussions in Appendix. B to address this question.
> - (b) Specifically, DORA constructed an additional MDP to estimate the Exploration-value as a generalized counter for count-based exploration, yet those count-based methods are orthogonal to reward shifting: in intrinsic reward methods, an agent must **first experience** a new $(s,a)$ pair before receiving a high intrinsic reward --- this is extremely hard with an arg-max style policy. On the other hand, with optimistic initialization, the rarely-visited $(s,a)$ pairs will naturally have higher $Q$-values **before experiencing** it --- as the frequently-visited pairs have updated their values with a negatively shifted reward. From such a perspective, reward shifting not only works by itself motivates exploratory behaviors but can also be seamlessly plugged into intrinsic reward methods to **encourage the first visitation** of new states. We’ve updated this difference to section 4.3.2.
> - (c) Moreover, we do not intend to use our experiments to show that reward shifting is beating other methods of online exploration or offline exploitation, but use those experiments to demonstrate the key insight of reward shifting is equivalent to different initializations that can generally be applied to value-based methods. Combining DORA with reward shifting is also a promising direction to gain further improvement.
> - (d) In Distributional RL literature [1-5], the distribution of the $Q$-value, rather than the mean scaler, is estimated. Distributional-RL focuses on stochastic reward mechanisms and smooths the temporal difference learning at a distributional level, and can be applied to risk-sensitive scenarios [6] where the worst-case performance can be controlled [7]. In those scenarios, systematic uncertainty is the crucial issue to address, whereas, in our work, we focus on deterministic transition dynamics and use OFU to tackle the epistemic uncertainty in the section of RRS.
> - (e) Besides [8, 9], several previous works discussed ensemble methods for exploration, both for discrete control [10] and for continuous control [11, 12]. However, in our work, we show that more exploratory behavior can emerge with the help of reward shifting under only a single $Q$-value network — as proof of the concept that negative reward shift is equivalent to optimistic initialization.
>
>
>
>
>
> ### Q3: More Challenging Environments/ additional Experiments
> - (a) We **added new experiments** on the **SparseHalfCheetah** introduced in VIME (a reward of +1 is provided only when the forward velocity is larger than 5 units per timestep) and **Humanoid** — a high dimensional continuous control task in the MuJoCo locomotion suite to verify the effectiveness of reward shifting in exploration.
> - (b) In SparseHalfCheetah, we find a negative reward shift leads to more explorative behavior and improves the learning efficiency while a positive reward shift hinders the learning efficiency.
> - (c) In Humanoid, we find using a reward shift can drastically improve the asymptotic performance by ~$+60\%$.
> - (d) Experiments are repeated with $8$ runs.
>
>
>
>
> ### Q4: Figures 2 and 3 I found kind of confusing.
> - We’ve updated and colorized the captions to make the presentation clearer.  We also updated the presentation and formally introduce the notions before the figures.

---

> ### Author Response · Authors · 2022-08-05
> **Dear Reviewer FqMf**
>
> Thanks again for reviewing our paper.
>
> We were curious if there should remain any other concerns on your part. We believe that we have addressed the concerns mentioned in your review, and are happy to clarify further should you remain unsatisfied.
>
> Best wishes,
> Paper #716 Authors

---

> > ### Comment · Reviewer_FqMf · 2022-08-05
> > **Please don't nag like this**
> >
> > I appreciate your enthusiasm to engage and receive feedback, but it takes time to review changes, and daily nag messages are annoying and do not help your case.
> >
> > Please see my comments on your changes attached to my rebuttal acknowledgement at the top.

---

### Official Review · Reviewer_wKjq · 2022-07-11

**Rating:** 4
**Confidence:** 5
**Soundness:** 2 fair
**Presentation:** 3 good
**Contribution:** 2 fair

**Summary:**

The paper first shows that reward shifting is equivalent to diversified Q-value network initialization in deep-RL. This kind of phenomenon can influence learning efficiency. Then, using the key insight, the paper presents three scenarios to show the benefit of reward shifting. Finally, the paper conducts experiments to prove the key insight in three scenarios.

**Questions:**

1. In lines 40-41, is there any theoretical proof of this conclusion? Can you make sure in most of the conditions, the key insight is true?
2. Why the baselines in 5.2 and 5.3 are different?
3. How about the total amount of compute between the baselines and the algorithms in this paper?
4. What are the advantages and disadvantages of the proposed method compared to similar exploration enhancement algorithms? Lack of description or theoretical comparisons.
5. Why does EnsemleTD3 perform worse than TD3 in Figure 5?
6. How can fairness be guaranteed by using only 3 Q networks in Figure 5?
7. There is no doubt that negative reward shifting will lead to more exploratory behavior, but is the Reward Shift approach better than Count-Based or other approaches? In Figure 1, Count-Based and epsilon-greedy can also be adjusted for exploration ability by hyperparameters, are these adjusted to be optimal in the experiment?

**Limitations:**

Yes.

**Strengths And Weaknesses:**

Strengths:
1. The paper shows that reward shifting is equivalent to diversified Q-value network initialization in deep-RL. This is the key point throughout the paper.
2. In 4.3.1, the paper proposes that Q-value can be written in the form of a linear combination.
3. The experiments compare many conditions in one environment.

Weakness:
1. The experiment results are not reliable enough. In Figure 4, the paper shows four kinds of Hopper environment and one kind of Walker environment. And the result in the Walker environment is not so obvious. More environments and more experiments in each environment are needed to compare. The experiment shown in Figure 6 has the same problem. In Figure 5, only 1e6 steps of environmental interactions are used, resulting in policies that have not yet converged in most tasks.
2. The theoretical proof is not good insufficient. We should be more concerned about how to find the balance of the conservative and optimistic exploration, rather than attempting different reward shifting settings for each environment.
3. The innovation of this paper requires further discussion. This paper is based on an intuitive idea to optimize multiple classes of methods such as offline and online. This paper treats each class of methods differently, and there is only one baseline algorithm for each class of methods, which makes me worry about the universality of this optimization.

---

> ### Author Response · Authors · 2022-08-02
> **Response to Reviewer wKjq [Part 2/2]**
>
>
> ### Q6: How about the total amount of computing between the baselines and the algorithms in this paper?
> - The hardware and computational time are provided in Appendix C. **As we have noted in Checklist 3(d).** In general, shifting the reward does not introduce further computation burden except in the continuous control tasks, our method of Random Reward Shift (RRS) requires two additional $Q$-value networks. In our PyTorch-based implementation, those additional networks can be easily implemented and optimized in a parallel manner, and the extra computational burden is equivalent to using a $\sqrt{3}$ times wider neural network during optimization. It is worth noting that RRS is computationally much cheaper than the Bootstrapped TD3, where additional policy networks are also needed.
>
>
>
> ### Q7: What are the advantages and disadvantages of the proposed method compared to similar exploration enhancement algorithms? Lack of description or theoretical comparisons.
> - We presented the difference between reward shifting and related works in Table 1, and discussed it in detail in Section 4.3.2. We've included an extended discussion on related works in Appendix E.
>
> ### Q8: Why does EnsemleTD3 perform worse than TD3 in Figure 5?
> - (a) First of all, the claim of ``EnsemleTD3 performs worse than TD3 in Figure 5’’ **is not true** — Ensemble TD3 only underperforms TD3 in one of the five environments.
> - (b) Ensembling policies in RL are not as trivial as in supervised learning: there is no fixed dataset for training a diverse set of models. Instead, policies need to interact with the environment and collect the replay buffer for its learning. For the pursuance of sample efficiency, we definitely can not let $K$ TD3 agents interact with the environments and update their policies in isolation. In our work, we implement this Ensemble TD3 by using multiple value networks with reward shift constant $0$, and randomly update the policy with one of those value networks, where instability can be introduced and hinder the performance. This set of experiments works as an ablation study to clarify the source of performance gain from RRS.
>
> ### Q9: How can fairness be guaranteed by using only 3 Q networks in Figure 5?
> - (a) In our experiments, 10 random seeds are used to demonstrate the statistical differences between performances. We provided experiments with other choices (i.e., 7 networks rather than 3) on the number of Q networks in Appendix C.2.
> - (b) Could the reviewer please explain more on this question for clarity?
>
>
> ### Q10: Is the Reward Shift approach better than Count-Based or other approaches?
> - Reward shift can be regarded as an extended version of the count-based exploration while the latter can not be applied to continuous state space. In our experiments, RND is a proxy of such count-based exploration. For a fair comparison, in all experiments, we use exactly the same hyper-parameters for DQN/RND and their combination with reward shifting.
>
> ---
> ### References
>
> [1] Fujimoto, Scott, Herke Hoof, and David Meger. "Addressing function approximation error in actor-critic methods." International conference on machine learning. PMLR, 2018.
>
> [2] Fujimoto, Scott, David Meger, and Doina Precup. "Off-policy deep reinforcement learning without exploration." International conference on machine learning. PMLR, 2019.
>
> [3] Plappert, Matthias, et al. "Multi-goal reinforcement learning: Challenging robotics environments and request for research." arXiv preprint arXiv:1802.09464 (2018).
>
> [4] Mnih, Volodymyr, et al. "Human-level control through deep reinforcement learning." nature 518.7540 (2015): 529-533.
>
> [5] Silver, David, et al. "Deterministic policy gradient algorithms." International conference on machine learning. PMLR, 2014.
>
> [6] Lillicrap, Timothy P., et al. "Continuous control with deep reinforcement learning." arXiv preprint arXiv:1509.02971 (2015).

---

> ### Author Response · Authors · 2022-08-02
> **Response to Reviewer wKjq [Part 1/2]**
>
>
> Thank you for your time and feedback. We would like to respond to each weakness and question in turn.
>
> ---
>
> ### Q1: Performance, Experiment results in Figure 4, Figure 5, Figure 6.
>
> - (a) Our experiment results in Figure 4 show that positive reward shifting consistently improves learning performance in terms of both computational efficiency and asymptotic performance. Importantly, in Walker, reward shifting (~ +3000) **drastically outperforms **CQL(~+1500) by **more than $100\%$**.
>
> - (b) For Figure 5, 1e6 timestep is a conventional choice of the deep RL literature. We follow the set-ups provided in our baseline [1]. The sample efficiency of RRS is 2.5x, 1.9x, 2.0x, 1.5x, and 1.6x higher than the TD3 baseline on the five tasks — **this is a 1.8x improvement on average.**
>
> - (c) In Figure 6, our experiment results demonstrate reward shifting improves performance on discrete action space exploration tasks. **It is clear in the results** that using negative reward shifting in DQN drastically outperforms the baseline, and applying negative reward shifting also clearly improves RND.
>
> ### Q2: Different settings.
> - "how to find the balance of the conservative and optimistic exploration’’ is the well-known dilemma for online RL, but is **not true** for the offline setting [2]. Also in deep-exploration tasks, the exploration is far more important than exploitation, as otherwise there is nothing that can be exploited at all (e.g., reward sparse tasks[3]). In our paper, we propose the reward shifting that is generally applicable to both online and offline settings — **this is not** ``attempting different reward shifting settings for each environment’’, but **demonstrating the efficacy of the underlying insight with a variety of tasks**.
>
>
> ### Q3: ``This paper treats each class of methods differently’’
> - Throughout our paper, we demonstrate the idea that
> `` *L40-41* A positive reward shifting leads to conservative exploitation, while a negative reward shifting leads to curiosity-driven exploration.’’.
> All settings, analyses, and experiments work together to prove the generality of the proposed insight *L40-41*.
>
> ### Q4:  ``there is only one baseline algorithm for each class of methods’’
> - **This is not true**. For offline-RL, we made a comparison based on both BCQ and CQL. For online exploration, we compared with DQN (with DQN + reward shifting) and RND (with RND + reward shifting). For continuous control, we compare against TD3, TD3 + ensemble, BootstrappedTD3.
>
> ### Q5: Why the baselines in 5.2 and 5.3 are different?
> - Because **benchmark algorithms for continuous control and discrete control are different**. DQN[4] can not be used for continuous control and the idea of learning a policy that selects the action with a maximum $Q$-value function is extended to continuous control by DPG[5], followed by an extended deep version DDPG[6], and then comes TD3[1] that further improves over DDPG.

---

> ### Author Response · Authors · 2022-08-05
> **Dear Reviewer wKjq**
>
> Thanks again for reviewing our paper.
>
> We were curious if there should remain any other concerns on your part. We believe that we have addressed the concerns mentioned in your review, and are happy to clarify further should you remain unsatisfied.
>
> Best wishes,
> Paper #716 Authors

---

> ### Author Response · Authors · 2022-08-08
> **Dear Reviewer wKjq**
>
> Given the limited time for the discussion period. We were curious if there should remain any other concerns on your part. **We believe that we have addressed the concerns mentioned in your review, and are happy to clarify further should you remain unsatisfied.**
>
> Best wishes,
>
> Paper #716 Authors

---

> ### Author Response · Authors · 2022-08-09
> **Please Let Us Know If There Are Leftover Concerns**
>
> Dear reviewer wKjq,
>
> Thank you again for your review and we believe we have addressed all your concerns. Considering the author-reviewer discussion period is coming to a close, please let us know if there are any other questions you may have.
>
> We believe reward shifting for value-based DRL is an important topic for research, with our work providing a significant contribution. If our response to your question was satisfactory, we would appreciate it enormously if you would consider raising your score.
>
> Best wishes,
>
> Authors #716

---

> > ### Comment · Reviewer_wKjq · 2022-08-10
> > **Thanks for the response**
> >
> > I thank the authors for the response, which helps me understand a bit more about the paper. The work is interesting, but it needs to be refined. I would like to maintain a borderline reject.

---

### Official Review · Reviewer_ZKxh · 2022-07-12

**Rating:** 6
**Confidence:** 3
**Soundness:** 3 good
**Presentation:** 3 good
**Contribution:** 3 good

**Summary:**

This paper proposes a reward shaping mechanism that trades of exploitation/exploration, by considering adding a negative vs. positive intrinsic constant leads to curiosity driven and conservative exploitation respectively, also they  study the effect of positive constant reward multiplication acts as a different learning rate. Based on this simple remark they study how offline and online RL can benefit from this shifting approach.




**Questions:**

Questions:
(See above)
. can the Q_UB-Q_LB provide a good uncertainty estimate locally?
Minor:
. missing meaning of each line if both fig.3 and fig. 2
. yellow in line166 (no yellow lines exist)


**Ethics Review Area:**

["I don’t know"]

**Limitations:**

- The paper studies how constant bias shift affect the value based learning agents as the constant is the same for all action/state space. In intrinsically motivated exploration this is not the case and it is a different one per state/action. Can the authors elaborate on insights of what this means and whether the current approach can make certain cases blind to exploration since the gap now is too big or small.
- how other OFU methods apply in this setting, such as Rmax.

**Strengths And Weaknesses:**

Strengths:
This paper is very well written and clear it studies a simple yet widely applicable problem in RL and curiosity driven approaches for exploration. It provides a derivation to how reward shifts affect the optimal value function, and their equivalence to different initialisations.
It tackles an important problem in exploration and provides interesting insights.

Weaknesses:
The paper does not show case the ranges under which this constant shifts work well, for too large gaps the the agent will never exploit or explore depending on sign?
There is also a trade-off between the multiplicative weight k and the bias? Can the authors clarify how these interact with each other?
How to best estimate a good b that provides a LB or UB? Can these parameters be learned?

---

> ### Author Response · Authors · 2022-08-02
> **Response to Reviewer ZKxh [Part 2/2]**
>
>
>
>
> ### Q4: Finding the best bias parameter.
> - (a) In general, it’s hard to find a constant that works for all environments as the reward scales in those environments are different. However, as discussed in A2, we can draw some heuristic guidance on the selection of such a hyper-parameter: for exploration tasks, using a constant with a large absolute value (compared to the scale of the reward of a specific task) will lead to exploratory behavior, thus more suitable for hard-exploration tasks. For exploitation tasks, using a relatively small constant can be safer and will not limit the extrapolation of policies.
> - (b) In practice, we find reward shifting outperforms baselines under a wide range of settings in conservative exploitation (cf., Figure 7) and consistently outperforms baselines for curiosity-driven exploration (in Maze tasks where deep exploration is necessary, cf., Figure 10 in appendix).
>
> ### Q5: scaling parameter $k$
> - As discussed in the paper, adjusting the scaling parameter $k$ does not change the optimal policy for discrete control (cf., Remark 1) and is equivalent to using a different learning rate for continuous control (cf., Remark 2). Therefore, we put our focus on reward shifting rather than scaling in this paper.
>
> ### Q6: Other OFU methods, Rmax
> - While Rmax focuses on model-based methods, we study reward shifting for the value-based model-free algorithms. Nonetheless, extending the basic idea of reward shifting to model-based RL — especially for uncertainty estimation based on different shifted reward mechanisms — is an interesting future direction.
>
> ### Q7: Universal bonus v.s. per state-action bonus?
> - (a) Initializing the function approximation with a universal constant to motivate exploration does not conflict with the fact that such a ‘bonus’ can be different during learning,
> - (b) We take the count-based exploration as an example and compare it with reward shifting: in count-based exploration, the intrinsic reward $r_{\mathrm{intrinsic}}(s,a) =  \frac{1}{K(s,a)}$, where $K(s,a)$ is the number of times the state-action pair $(s,a)$ has been visited, is added to the original reward function, the values of those intrinsic reward for different $(s,a)$ are the same at beginning, but varies during learning as different $(s,a)$-pairs are equally visited. In reward shifting, the value estimation for frequently-visited $(s,a)$-pairs is more precise (e.g., with MC estimation), and much lower than the rarely-visited pairs whose values are still close to the optimistic initialization.
> - (c) Different from count-based exploration where different (and importantly, **non-static**) bonus values are explicitly added to the reward function (hence, the aleatoric uncertainty of the value estimation is increased during learning as the bonus value varies), in reward shifting, such an intrinsic motivation is implicitly implemented by (**static**) optimistic initialization — the underlying optimal $Q$-value function is static during learning.
> Moreover, it is worth noting that count-based exploration can not be applied to continuous scenarios while reward shifting is generally applicable.
>
>
> ----
> ### References
>
> [1] Lakshminarayanan, Balaji, Alexander Pritzel, and Charles Blundell. "Simple and scalable predictive uncertainty estimation using deep ensembles." Advances in neural information processing systems 30 (2017).
>
> [2] Osband, Ian, et al. "Deep exploration via bootstrapped DQN." Advances in neural information processing systems 29 (2016).

---

> ### Author Response · Authors · 2022-08-02
> **Response to Reviewer ZKxh [Part 1/2]**
>
> Thank you for your comments and questions. We would like to respond to your questions and weaknesses.
>
> -----
>
> ### Q1: Update Figure 2 and 3:
>
>  - We’ve updated the captions in figure 2, 3 and colorized different notions for clarity.
>
> ### Q2. Local Uncertainty Estimators
>
> - (a) Yes, combining multiple neural networks for uncertainty estimation is a well-established method [1], and has been explored in the context of RL for discrete control [2].
>
> - (b) In this work, applying multiple reward shifting constant leads to multiple value functions, those different value functions can be used to estimate the local uncertainty by $v(s,a) = \mathbb{E}[(Q_i(s,a) - \mathbb{E}[Q(s,a)])^2]$.
>
> ### Q3: Under what range will the method fail?
>
> - (a) Our experiments have shown that for conservative exploration tasks, reward shifting may perform poorer than the baselines when the shifting constant is too large (e.g., shifting constant = 50, which is approximately 10 - 20 times larger than the maximal step-wise reward of the environment. cf. Figure 7.), as it can be over-pessimistic on extrapolations and thus limits the generalization ability of policies. For optimistic exploration tasks, reward shifting also may perform pooer than the baselines when the shifting constant is too large (e.g., shifting constant = -10, which is more than 10 times larger than the potential episodic return for navigation tasks. cf. Figure 10.).
>
> - (b) We further demonstrate the benefits of reward shifting for deep-exploration tasks and the on-par performance of reward shifting on easier tasks that do not require deep exploration. **The results are updated in Figure 11 in Appendix D.1**.
> We experiment on the maze environment and change the size of the maze to vary from 2 to 20, denoted as S2, S5, S10, S15, and S20, separately. We find in easy tasks (S2, S5, S10), both count-based exploration and reward shifting perform similarly to the $\epsilon$-greedy exploration, while on challenging tasks (S15, S20), more explorative behavior encouraged by reward shifting and count-based exploration is important for efficient learning.

---

> ### Author Response · Authors · 2022-08-08
> **Dear Reviewer ZKxh**
>
> Thanks again for reviewing our paper.
>
> We were curious if there should remain any other concerns on your part. We are happy to clarify further should you remain unsatisfied.
>
> Best wishes,
>
> Paper #716 Authors

---

### Author Response · Authors · 2022-08-03
**General response to all reviewers**

We would like to thank all reviewers for their time and useful feedback for improving the paper. We have responded to each reviewer separately, but would like to summarize our changes here to streamline potential discussion.

---
During the phase1 rebuttal, we've:

- 1. [**Presentation**] updated our paper and appendix and re-wrote some sections to improve the presentation, and marked the re-phrased parts with blue text, including but not limited to:
  - In the related work section, we include discussions on the works mentioned by the reviewer, and updated the phrasing in Table 1. We added extended discussions on related work at Appendix E.
  - In the method sections, we updated the presentation and formally introduce the notions before figures. We also updated the captions to make it clearer.
  - We’ve entirely rephrased the presentation in section 4.3.2 and section 5.3.

- 2. [**Experiments**] added three sets of new experiments in Appendix D.
  - In Appendix D.1, we experiment on the maze tasks with different size to show that reward shifting improves exploration in hard-exploration tasks, and does not decrease learning efficiency in easier tasks.
  - In Appendix D.2, we experiment on SparseHalfCheetah and Humanoid to show reward shifting can be applied to hard continuous control tasks.

---

Here please find the link to the updated appendix (for additional experiments in Appendix D; and extended discussions on related work in Appendix E.)
https://openreview.net/attachment?id=iCxRsZcVVAH&name=supplementary_material

---

### Author Response · Authors · 2022-08-04
**Dear Reviewers**

We are sincerely grateful for your time and energy in the review process.

In light of our responses and updated draft uploaded on Aug.2, we would appreciate it if the reviewer could kindly let us know of any leftover concerns. We would be happy to do our utmost to address them.

Thank you!
Paper 716 Authors

---

### Author Response · Authors · 2022-08-09
**Summary of Revision Changes**

We would like to thank all reviewers for their time, generous comments, and suggestions for improving the paper.

Besides the responses provided on a point-by-point basis, we have also updated the manuscript/supplementary to improve the paper. We wish to summarize the earlier updates to the experimental evaluations and manuscript/supplementary uploaded.

---
> ### New experiments


 1. **More Challenging Environments to Demonstrate the Effectiveness of Reward Shifting:** We additionally experiment on four continuous control tasks including locomotion and robotics (HalfCheetah-SparseReward, Humanoid, FetchReach, FetchPush) to highlight the general applicability of reward shifting. See Figure 12, Figure 13 in Appendix D.2, Appendix D.3 : (https://openreview.net/attachmentid=iCxRsZcVVAH&name=supplementary_material)

 2. **Less Challenging Environments to Show Reward Shifting is not Over-optimistic:** We adjusted the difficulty in the maze navigation task we used in Figure 1 by varying the distance between the starting point and the goal point, and used exactly the same reward shifting to show even in simple environments, explorative behaviors introduced by reward shifting will not step too far in exploration and the learning efficiency can still be guaranteed. See Figure 11 in Appendix D.1 (https://openreview.net/attachment?id=iCxRsZcVVAH&name=supplementary_material)


---
> ### Summary of updates to manuscript & supplementary


We've highlighted the changes made to our manuscript and supplementary material in different colors (blue for the phase-I rebuttal and orange for the phase-II rebuttal).

**Clarifications**
- We updated Table 1, to highlight the flexibility of reward shifting.
- We updated the presentation in the related work section, to better differentiate reward shifting from previous works.
- We updated the captions for Figure2 and Figure3, clarified the notions, and colorized the curves.
- We rephrased the method and experiment sections on reward shifting for value-based curiosity-driven exploration.
- We did multiple passes on writing for a clearer presentation.

**Extended Related Work**
- Extended discussions on count-based exploration methods are added to Appendix E.1
- Discussions on ensembles and distributional RL are added to Appendix E.2
- Discussions on model-based exploration and uncertainty estimation are added to Appendix E.3


---

With our responses and paper updates, we hope that we have addressed the reviewers' concerns. Please let us know if there were any further questions or comments. We are eager to do our utmost to address them!

Thank you for your kind consideration :)

Paper 716 Authors

---

### Meta-Review · Area_Chair_PoiR · 2022-08-25

**Recommendation:** Accept
**Confidence:** Less certain

**Metareview:**

This paper proposes a simple but general way to improve exploration in RL based on the equivalence between reward shifting and the initialization of value function. The paper shows that it is straightforward to implement conservative exploitation and curiosity-driven exploration based on this idea. The results on a variety of offline/online RL settings show that a properly initialized value function (i.e., shifted reward) can achieve a better exploration/exploitation and improve the performance of existing RL algorithms as a result.

In general, most of the reviewers found that the proposed method and the results are quite interesting enough to be presented at NeurIPS. While some reviewers had a concern about the lack of challenging tasks, the authors addressed it with updated results in the appendix. The only reviewer with a negative score did not respond during the discussion period. Thus, I recommend this paper to be accepted. In the meantime, there are still remaining (minor) concerns about the presentation of the paper (e.g., grammar, lengthy description of a simple idea, etc) and the lack of discussion on limitations and related work. I highly recommend the authors to improve them by reorganizing the paper (e.g., omit some details and move some important discussion from the appendix to the main text) for the camera-ready version.

**Award:**

No

---

### Decision · Program_Chairs · 2022-09-14

Accept